# Sildenafil Citrate Enhances Renal Organogenesis Following Metanephroi Allotransplantation into Non-Immunosuppressed Hosts

**DOI:** 10.3390/jcm11113068

**Published:** 2022-05-29

**Authors:** Ximo Garcia-Dominguez, César D. Vera-Donoso, Eric Lopez-Moncholi, Victoria Moreno-Manzano, José S. Vicente, Francisco Marco-Jiménez

**Affiliations:** 1Laboratory of Biotechnology of Reproduction, Institute for Animal Science and Technology (ICTA), Universitat Politècnica de València, 46022 Valencia, Spain; ximo.garciadominguez@gmail.com (X.G.-D.); jvicent@dca.upv.es (J.S.V.); 2Servicio de Urología, Hospital Universitari i Politècnic La Fe, 46026 Valencia, Spain; cdveradonoso@gmail.com; 3Neuronal and Tissue Regeneration Laboratory, Centro de Investigación Príncipe Felipe, 46012 Valencia, Spain; elopezm@cipf.es (E.L.-M.); vmorenom@cipf.es (V.M.-M.)

**Keywords:** kidney, metanephros, organogenesis, transplantation, regenerative medicine

## Abstract

In order to harness the potential of metanephroi allotransplantation to the generation of a functional kidney graft on demand, we must achieve further growth post-transplantation. Sildenafil citrate (SC) is widely known as a useful inductor of angiogenesis, offering renoprotective properties due to its anti-inflammatory, antifibrotic, and antiapoptotic effects. Here, we performed a laparoscopic metanephroi allotransplantation after embedding sildenafil citrate into the retroperitoneal fat of non-immunosuppressed adult rabbit hosts. Histology and histomorphometry were used to examine the morphofunctional changes in new kidneys 21 days post-transplantation. Immunofluorescence of E-cadherin and renin and erythropoietin gene expression were used to assess the tubule integrity and endocrine functionality. After the metanephroi were embedded in a 10 µM SC solution, the new kidneys’ weights become increased significantly. The E-cadherin expression together with the renin and erythropoietin gene expression revealed its functionality, while histological mature glomeruli and hydronephrosis proved the new kidneys’ excretory function. Thus, we have described a procedure through the use of SC that improves the outcomes after a metanephroi transplantation. This study gives hope to a pathway that could offer a handsome opportunity to overcome the kidney shortage.

## 1. Introduction

Currently, renal diseases affect epidemic numbers of people worldwide and have continued to escalate in their prevalence globally in recent years [1]. Kidney organs are responsible for vital functions, including the excretion of metabolic wastes and toxins, body fluid regulation, and the endocrine control of the blood pressure and erythrocyte maturation. Hence, when renal degenerative processes end in an organ failure, organ transplantation becomes the ideal method for restoring full physiological organ function [2]. However, the unavailability of suitable organs for transplantation forces end-stage renal patients to decide to take dialysis treatment or die. In Spain, the world’s leading country in organ donation for 28 consecutive years [3], only 3423 kidney transplantations were performed in 2019, instead of the 7356 that were necessary according to the waiting list [4]. In 2018, 40% of US patients listed for a kidney transplant were still waiting since 2015, and 34,591 patients were removed from the list due to death or decline in medical condition [5,6]. Therefore, the required organs do not arrive in time for all the patients, and 5–10% of patients die on the waiting list every year [7]. In this precarious situation, emerging technologies in the field of regenerative medicine seek to address the limitations of current treatment strategies, exploring new frontiers. The common idea is to generate kidney grafts on demand to function as native kidneys, based on strategies ranging from stem cell therapy, blastocyst complementation, decellularization-recellularization, or 3D bioprinting [8,9]. However, the kidney is one of the most challenging organs for de novo formation due to its complex architecture and composition, containing numerous highly specialized and differentiated cell types [8,10]. Therefore, to date, cell therapies with individual cells are far from achieving fully functional transplantable renal grafts. As a promising solution for these limitations, the xenotransplantation of embryonic kidneys showed that, if this intact renal primordium was transplanted into adult non-immunosuppressed hosts, then it could mature as if they had not been extracted from the embryo, with a significantly reduced immune response in the hosts [11,12]. These embryonic kidneys (metanephroi) are able to attract the formation of a vascular system from the host, undergoing maturation and exhibiting excretory and endocrine functional properties [11,12,13,14,15,16,17,18,19,20]. Glomerular filtration in developing metanephroi was demonstrated firstly in the 1990s [21,22]. Today, metanephroi transplantation remains a promise to treat the renal injury, as metanephroi have been successfully transplanted across concordant and highly disparate xenogeneic barriers [12,20,23,24]. Specifically, Dekel et al. transplanted human and porcine metanephroi into mice, obtaining kidney structures that produce urine [12]. These findings suggest that if embryonic organs are retrieved from pathogen-free animals [9,25], this source could provide an unlimited and elective supply of organs for clinical transplantation.

However, allowing the new kidney to grow larger and sustain life in the long-term is a remaining obstacle to guarantee the feasibility of this strategy for clinical application [18,19,26,27,28,29]. Using combinations of growth factors, Hammerman’s group have achieved rates of clearance in transplanted metanephroi almost 300 times those measured without any treatment [19]. These values were approximately 6% of the clearance achieved by a normal kidney [27,30], which approximates a renal function level that would be expected to preserve life [30]. Therefore, it is of special importance to investigate whether growth-promoting factors could be used to enhance the growth and function of developing metanephroi. Sildenafil citrate (SC) is a well-known drug used to treat pulmonary hypertension and male erectile dysfunction due to its vasodilatory effect. SC up-regulates cGMP, nitric oxide, and angiogenic systems, causing angiogenesis and renoprotective effects through anti-inflammatory, anti-oxidant, and anti-apoptotic mechanisms [31,32,33,34,35]. SC has demonstrated beneficial properties as a preconditioning or protective drug during kidney transplantation [36,37]. Besides, SC enhances the cartilage graft viability and survival, which is highly dependent on the vascularized host bed, oxygenation of local tissue, and the patient’s current systemic status [34]. To some extent, the survival and development of the avascular metanephroi could depend on the same variables, being crucial in its connection to the host vascular system. Therefore, this study was conceived to explore if SC addition during the metanephroi transplantation improves its development. 

## 2. Materials and Methods

All chemicals, unless otherwise stated, were reagent-grade and purchased from Sigma-Aldrich Química SA (Alcobendas, Madrid, Spain). 

### 2.1. Ethical Statements

The study was approved by the Universitat Politècnica de València Ethical Committee (Code: 2015/VSC/PEA/00170). The study followed the Directive 2010/63/EU EEC guidelines. Experimental protocols were conducted under the supervision of the animal welfare committee in charge of this animal facility. An authorisation certificate issued by the Valencian governmental administration to experiment on animals is held by X. GD (code: 2815), F. MJ (code: 2273), and JS. V (code: 0690).

### 2.2. Experimental Design

New Zealand rabbits were used for the experiment. Metanephroi were recovered from 15-day-old (E15) embryos. Then, the metanephroi were placed in 5 µL drops containing one of the following SC concentrations: 0 µM (0 SC; untreated group), 10 µM (10 SC), and 30 µM (30 SC). After that, metanephroi were laparoscopically transplanted into non-immunosuppressed adult hosts (5 months). Total white blood cells and lymphocytes were estimated to identify any immunological response. After 21 days, the transplantation efficiency (recovery rate: kidneys recovered/metanephroi transplanted), nascent kidney growth (weight), its excretory function (histology, histomorphometry, and hydronephrosis), the tubule integrity (E-Cadherin), and its endocrine function (mRNA) were assessed. Kidneys originated from neonatal rabbits (1 week-old, coeval with metanephroi age) were used as control. The experimental design is summarized in Figure 1.

### 2.3. Metanephroi Recovery and Transplantation

Metanephroi were carefully dissected from E15 rabbit embryos under a dissecting microscope. One embryo was fixed directly for histological examination. Just before its allotransplantation, metanephroi were deposited in 5-μL droplets of phosphate-buffered saline (PBS) solution containing SC concentrations of 0 µM (untreated group), 10 µM, or 30 µM. All manipulations were performed at room temperature (25 ± 1 °C). The allotransplantation was performed using a minimally invasive laparoscopic technique, as previously described [38], within 45 min after the metanephroi were retrieved. Briefly, animals were placed on an operating table in a vertical position (head down at a 45-degree angle). Only one endoscope trocar was inserted into the abdominal cavity. Then, an epidural needle was inserted into the inguinal region. After a renal vessel was identified in the retroperitoneal fat, a hole (as a pouch) was performed adjacent to the vessel. Then, each metanephros was aspirated with 5 µL of each SC solution (0 SC, 10 SC, or 30 SC) in an epidural catheter (Vygon corporate, Paterna, Valencia, Spain), introduced into the inguinal region through an epidural needle, and deposited (transplanted) into the pouch previously created. Between 5 to 9 metanephroi were transplanted in each host (one metanephros per hole). A total of seven adult rabbits were used as hosts in three sessions, without immunosuppressive therapy (2, 2 and 3 hosts for 0, 10 and 30 SC, respectively). Anaesthesia, analgesia, and the postoperative care were performed as we previously described for laparoscopic procedures [39]. 

### 2.4. Determination of Peripheral White Blood Cells

Before the transplantation (day 0), a blood sample of each host (*n* = 7) was collected from the central ear artery and dispensed into an EDTA-coated tube (Deltalab S.L., Barcelona, Spain). Then, basal levels of total white blood cells and lymphocytes were estimated at most 10 min after blood collection, using an automated veterinary haematology analyser (MS 4e automated cell counter, MeletSchloesing Laboratories, Osny, France) and according to the manufacturer’s instructions. After metanephroi transplantation, two blood extractions were analysed weekly along the experiment to detect significant variations of the total white blood cell and lymphocyte populations. 

### 2.5. Metanephroi Development and Histomorphometry of the Renal Corpuscle

All animals were euthanized 3 weeks after transplantation, retrieving all the new kidneys developed to annotate the recovery rate (recovered kidneys/transplanted metanephroi). Then, renal structures were weighted, fixed in formaldehyde solution, and embedded in paraffin wax for histological analysis. Samples for histology were cut into 5-μm sections and stained with haematoxylin and eosin. The stained sections were observed with light microscopy for histological and histomorphometric examination. To measure histomorphometric parameters, a minimum of 25 renal corpuscles and glomeruli were evaluated (area and perimeter) for each experimental group. Photomicrographs were taken at a total magnification of ×400. Measurements were determined using ImageJ software (public domain http://rsb.info.nih.gov/ij/, accessed 01 April 2022). In addition, the glomerular tuft cellularity was estimated by counting the total number of nuclei of each glomerulus. Kidneys originating from a 5-week-old rabbit (coeval with the metanephroi age) were used as controls.

### 2.6. Tubule Integrity by Targeting E-Cadherin 

Immunofluorescence for paraffin-embedded kidney 5-μm sections required prior de-waxing, rehydration, and antigen retrieval (immersion in tris-EDTA buffer (10 mM Tris, pH 9.0) for 25 min at 97 °C) steps. Then, samples were incubated with blocking solution (5% horse serum, 10% fetal bovine serum in phosphate buffer solution with 0.1% Triton X-100) for 1 h at room temperature and incubated with the primary antibody mouse anti-E-Caherin (Cat. C20820; BD bioscience, Franklin Lakes, NJ, USA) overnight in a humidified chamber at 4 °C. After washing, the sample was incubated with the secondary antibody (Alexa-Fluor 555; 1:400; Invitrogen, Waltham, MA, USA) at room temperature for 2 h. All cells were counterstained by incubation with 4,6-diamidino-2-phenylindole dihydrochloride (DAPI; Invitrogen). After a final wash, the sections were evaluated by using the Apotome Inverted Fluorescence Microscope (Zeiss, Jena, Germany). Consistent exposures were applied for all images.

### 2.7. Renin and Erythropoietin mRNA Gene Expression 

After euthanasia, developed metanephroi samples were obtained by retrieving biopsies randomly from different sites. Immediately, samples were washed with PBS to remove blood remnants and stored in RNA-later (Ambion Inc., Huntingdon, UK) at −20 °C until the analysis. Five samples were analysed in each experimental group (control, 0 SC, 10 SC, and 30 SC). Host kidneys (under the same physiological environment as nascent kidneys) were used as the control. RNA was extracted with a Dynabeads kit (Invitrogen Life Technology) according to the manufacturer’s instructions and treated with DNase I to eliminate genomic DNA contamination. Then, reverse transcription was carried out using a Reverse Transcriptase Quantitect kit (Qiagen, Hilden, Germany). Real-time quantitative PCR (RT-qPCR) reactions were conducted in an Applied Biosystems 7500 (Applied Biosystems, Foster City, CA, USA). Every RT-qPCR was performed from 5 µL of diluted 1:40 cDNA template, 250 nM of forward and reverse primers (Table 1), and 10 µL of PowerSYBR Green PCR Master Mix (Fermentas GMBH, Madrid, Spain) in a final volume of 20 µL. The PCR protocol included an initial step of 50 °C (2 min), followed by 95 °C (10 min), and 42 cycles of 95 °C (15 s) and 60 °C (60 s). After RT-qPCR, a melting curve analysis was performed by slowly increasing the temperature from 65 °C to 95 °C, with the continuous recording of changes in fluorescent emission intensity. The amplification products were confirmed by SYBR Green-stained 2% agarose gel electrophoresis in 1X Bionic buffer. Serial dilutions of the cDNA pool made from several samples were conducted to assess RT-qPCR efficiency. A ΔΔCt method adjusted for RT-qPCR efficiency was used [40], employing the geometric average of glyceraldehyde-3-phosphate dehydrogenase (GAPDH) as the housekeeping normalization factor [41]. Relative expression of the cDNA pool from various samples was used as the calibrator to normalise all samples within one RT-qPCR run or between several runs.

### 2.8. Statistical Analyses

Differences in the recovery rates between groups were assessed using a probit link model with binomial error distribution, including the experimental group as a fixed effect. Variations in the peripheral blood cells were evaluated using a general linear model (GLM), including the day post-transplant as a fixed factor. The experimental group was non-significant and was removed from the model. The new kidney weight, hitomorphometric measures (area and perimeter), and the glomerular tuft cellularity were compared using a GLM, including the experimental group as a fixed effect and a replicate as a random factor. The replicate was non-significant and was removed from the model. Data of relative mRNA abundance were normalized by a Napierian logarithm transformation and evaluated using a GLM as previously described. Data were expressed as least square means ± standard error of means. Differences of *p* ≤ 0.05 were considered significant. All statistical analyses were performed with SPSS 21.0 software package (SPSS Inc., Chicago, IL, USA).

## 3. Results

### 3.1. Allotransplanted Metanephroi Form Adult Organs

Three New Zealand white rabbits were used as embryo donors, obtaining a total of 28 E15 embryos. Metanephroi were carefully micro-dissected and transplanted into seven adult non-immunosuppressed hosts (5-month-old animals). A total of 49 whole metanephroi were allotransplanted: 15 in 0 SC (untreated group), 17 in 10 SC, and 17 in 30 SC. The peripheral circulating white blood cell count (total, lymphocytes, monocytes, and eosinophils) remained unchanged after allotransplantation (Figure 2).

Twenty-one days after transplantation, we observed that the transplanted metanephroi grew and promoted angiogenesis (Figure 3). Metanephroi treated with SC exhibited a macroscopic view of deeper vascular integration than the untreated ones (Figure 3). A similar recovery rate was observed for all the groups: 10/15 (67%), 9/17 (53%), and 8/17 (47%) for 0 SC, 10 SC, and 30 SC groups, respectively.

### 3.2. Comparative Renal Weight and Histomorphometry Study

Significant increase in developing kidney weight was observed for the 10 SC group (0.13 ± 0.021 g), compared with the 30 SC (0.07 ± 0.025 g) and the 0 SC (0.08 ± 0.020 g) groups. All nascent kidney weights were lower than the control samples (0.74 ± 0.028 g), independently of the experimental group (*p* < 0.05, Figure 4). 

All nascent kidneys became hydronephrotic, demonstrating its excretory function. Concordantly, in all the groups, metanephroi underwent differentiation and developed new kidney graft explants with histologically mature glomeruli (Figure 5), whose histomorphometric analysis is shown in Table 2. 

The histomorphometric data showed that all the metanephroi-developed kidneys exhibited higher renal corpuscle values (area and perimeter) than the control samples, demonstrating the hydronephrotic state of the former and its filtering activity. Moreover, both renal corpuscle and glomerulus measurements were increased in the 10 SC and 30 SC groups compared to the untreated one, suggesting that SC increased both the capillary dilatation and glomerular filtration. Tuft cell density in the 10 SC and 30 SC groups’ developed metanephroi were also higher than in the untreated one, and similar to the control samples, they showed a trophic effect of the SC on the glomeruli development. 

### 3.3. Tubule Integrity in the New Kidneys

Immunofluorescence assay results showed that E- cadherin was highly expressed in new kidneys (Figure 6B–D). E-cadherin expression reflected the reduced tubule size in the 30 SC group (Figure 6D).

### 3.4. Glomerular Renin and Erythropoietin Production in the New Kidneys

Renin and erythropoietin mRNA levels were similar between metanephroi-developed kidneys and the control samples (host’s kidney), regardless of the experimental group (Figure 7). 

## 4. Discussion

Herein, we have established a protocol to obtain an enlarged kidney after the use of SC during metanephroi transplantation. Moreover, our findings revealed that SC also positively affects glomerular development and function without affecting the tubule integrity and the endocrine properties. Besides, our data provide that there are no substantial changes in the magnitude of lymphocyte population, regardless of the SC concentration used. Altogether, our results represent firm evidence of the SC trophic effects for metanephroi development. Looking for the strategies that allow obtaining life-sustaining renal structures, SC‘s use becomes a potential factor towards the clinical translation of metanephroi transplantation therapy.

Severely damaged kidneys possess a limited regenerative potential, and therapeutic interventions are not sufficient to restore renal function in patients with ESRD, turning transplantation into the ideal method to restore full physiological functions [8,42]. Given the graft shortage, either from living or deceased donors, well recognized by WHO [43], some regenerative and bioengineering strategies are trying to generate kidney grafts on demand [8,9,44]. Metanephroi transplantation remains one of the most promising approaches, but obtaining larger and life-sustaining renal structures after its transplantation remains an obstacle for its therapeutic potential [19,26,27,28,29]. In this sense, we evaluated the SC effect during a metanephroi transplantation on their in vivo development. Previously, Rostaing et al. demonstrated that SC exerts a dilatation of glomerular afferent arterioles, promoting an increase in the filtration process [45]. Likewise, our histomorphometric data showed that SC increases the area and perimeter of the glomerulus and the renal corpuscle. If SC promotes glomerular filtration in nascent kidneys, they must accumulate more filtrate in the bowman’s space due to the lack of a urine-excretion channel (hydronephrosis). Taking into account that SC acts as a potent inductor of VEGF (vascular endothelial growth factor) release [46,47,48], our results support those of Hammerman’s group, who indicated that urine volumes were increased significantly in VEGF-treated metanephroi [27]. However, although VEGF treatment did not affect the weights of transplanted metanephroi [27], SC treatment allows us to obtain larger renal structures. As a possible explanation, it has been proven that low SC doses resulted in more angiogenic responses than those produced by a saturating VEGF concentration [46]. Besides, SC has been previously proposed as a helpful agent in instances where neo-vascularization is desired [46], enhancing graft viability [34]. On the other hand, the renoprotective effects attributable to SC could retard hydronephrosis-related damages [31,32,33,34,35], allowing metanephroi growth for longer. However, this overgrowth appeared in the 10 SC group but not in the 30 SC one. As hydronephrosis turns metanephroi not viable [18], maybe high SC doses could accelerate glomerular filtration, exacerbating a hydronephrotic state that arrests the metanephroi growth earlier. Similar results were observed by Yardimci et al. [49], which allow us to speculate that metanephroi development could be better in the 30 SC group before urine production was started. Therefore, the use of 10 SC allows us to recover larger renal structures with a higher degree of glomerular development and function. Moreover, developed metanephroi showed glomerular filtration activity and endocrine functions, which are consistent with the previous literature [12,14,15,17,18,19,27]. We have identified the expression of E-cadherin in the renal tubules of the new kidneys. E-cadherin is a key adherent protein in the formation of adhesion junctions, which are critical to tubule integrity in normal kidneys [50]. In addition, the renin and erythropoietin gene expression levels were similar between new and host kidneys. Interestingly, this study was consistent with previous reports showing that harvesting metanephroi at the optimal age avoids the immunological response from hosts [38,51]. Compared to coeval native organs, developed metanephroi reach a diminished renal mass that could incur potential life-sustaining limitations. However, it has been demonstrated that the survival time of anephric recipients transplanted with metanephroi is proportional to the renal mass developed [52]. This concept is similar to that used in kidney transplantation from paediatric donors, in which both kidneys are transplanted in bloc into adult recipients to guarantee acceptable glomerular filtration rates [53,54]. This strategy should be combined with those designed to avoid hydronephrosis [18], or others developed in our group, such as the minimally invasive laparoscopic transplantation procedure [38,55], effective banking protocols [56,57,58], and, now, the use of SC. All these strategies, in conjunction, could constitute a path increasingly consolidated by which metanephroi xenotransplantation could provide transplantable renal grafts to treat patients with ESRD.

## 5. Conclusions

In conclusion, we have reported a procedure based on SC’s addition during the metanephroi transplantation that promotes the nascent kidneys’ growth and glomerular filtration. This treatment enhances the glomerular developmental degree without compromising either the endocrine activity or the new renal structures’ immunological silence. This study gives hope to a pathway that could offer a handsome opportunity to overcome the kidney shortage.

## Figures and Tables

**Figure 1 jcm-11-03068-f001:**
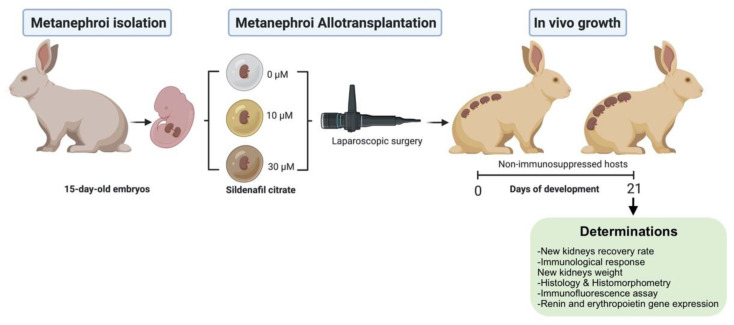
Experimental design. Metanephroi were recovered, embedded in 5 µL drops of sildenafil citrate (0, 10, 30 µM), and laparoscopically transplanted into non-immunosuppressed hosts. The immunological response, recovery rate (kidneys recovered/metanephroi transplanted), and the new kidneys weight and function (excretory and endocrine) were assessed (figure created with BioRender.com).

**Figure 2 jcm-11-03068-f002:**
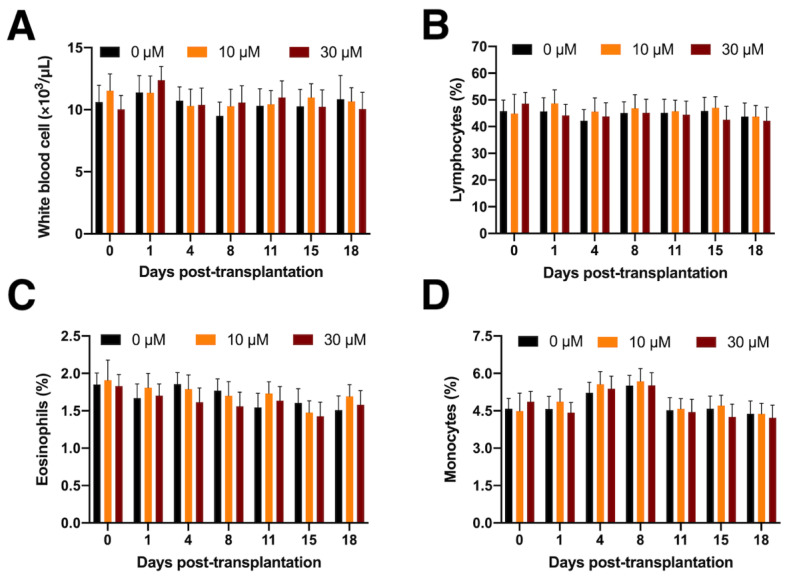
White blood cell counts after metanephroi allotransplantation: (**A**) total white blood cells; (**B**) lymphocytes; (**C**) eosinophils; (**D**) monocytes.

**Figure 3 jcm-11-03068-f003:**
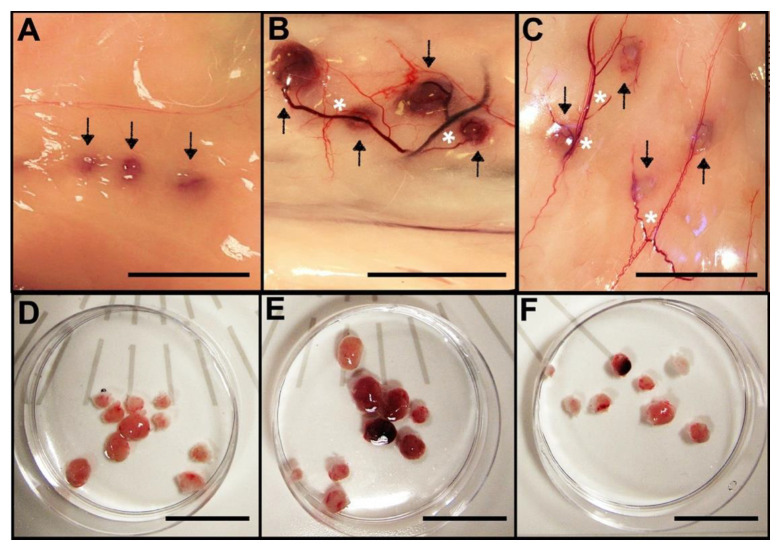
Development of new kidney recovery 21 days after metanephroi allotransplantation with or without sildenafil citrate (SC). (**A**) Nascent kidney from metanephroi without SC treatment. Arrows indicate the growing metanephroi. (**B**) Developing kidney from metanephroi treated with the 10 µM SC solution. Asterisk indicates neoangiogenesis. Arrows indicate the growing metanephroi. (**C**) Developing kidney from metanephroi treated with the 30 µM SC solution. Asterisk indicates neoangiogenesis. Arrows indicate the growing metanephroi. (**D**–**F**) New kidneys recovered from transplanted metanephroi (**D**) without SC treatment, (**E**) with the 10 µM SC solution, and (**F**) with the 30 µM SC solution. Scale bars: 2 cm.

**Figure 4 jcm-11-03068-f004:**
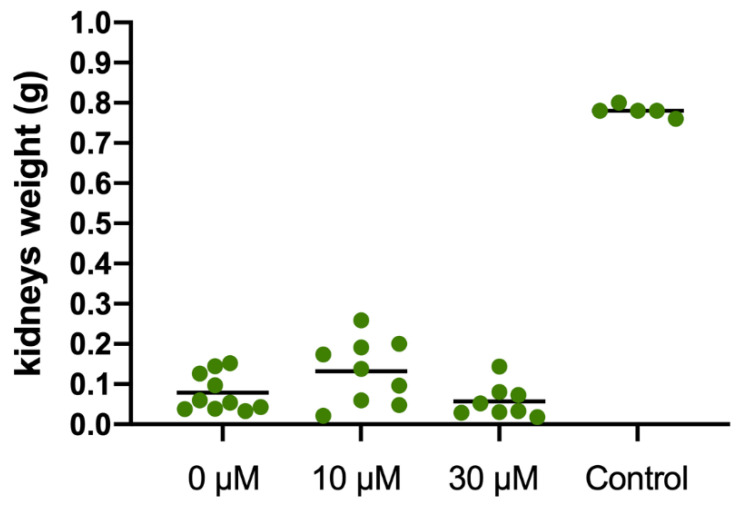
Kidney weight recovery 21 days after metanephroi allotransplantation embedding in sildenafil citrate (0, 10 and 30 µM). The control kidney originated from a neonatal rabbit (1 week-old).

**Figure 5 jcm-11-03068-f005:**
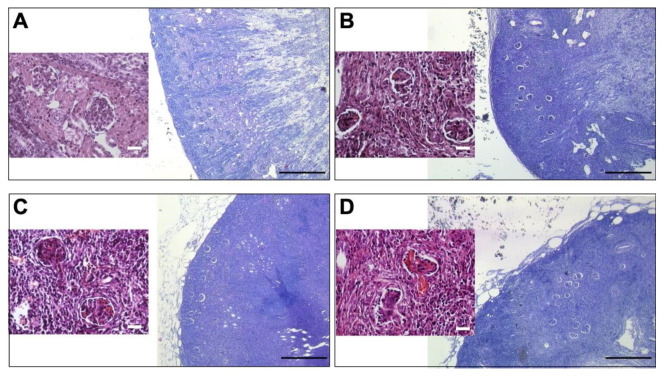
Histology of developing kidneys with or without sildenafil citrate. (**A**) Micrograph showing the glomerulus details (original magnification, ×400) and the outer renal cortex (original magnification, ×40) of the control kidney originating from a neonatal rabbit (1 week-old). (**B**–**D**) Micrograph showing the glomerulus details (original magnification, ×400) and the outer renal cortex (original magnification, x40) of a new kidney after metanephroi transplantation: (**B**) without sildenafil citrate (SC) treatment, (**C**) with a 10 µM SC solution, and (**D**) with a 30 µM SC solution.

**Figure 6 jcm-11-03068-f006:**
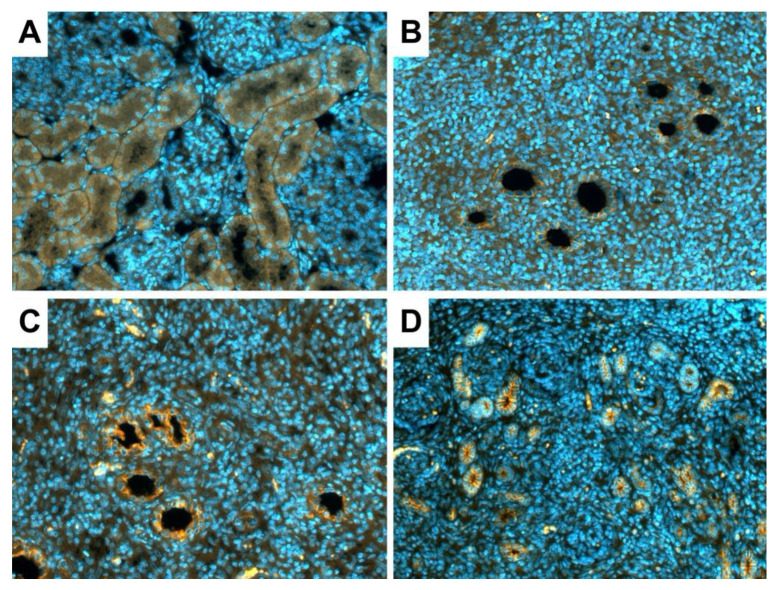
Immunofluorescence and confocal microscopy assay for the detection of tubular integrity marker E-Cadherin. (**A**) Micrograph showing tubular integrity of the control kidney originating from a neonatal rabbit (1 week-old). (**B**–**D**) Micrograph tubular integrity of the new kidney after metanephroi transplantation: (**B**) without sildenafil citrate (SC) treatment, (**C**) with a 10-µM SC solution, and (**D**) with a 30-µM SC solution. Original view: × 20.

**Figure 7 jcm-11-03068-f007:**
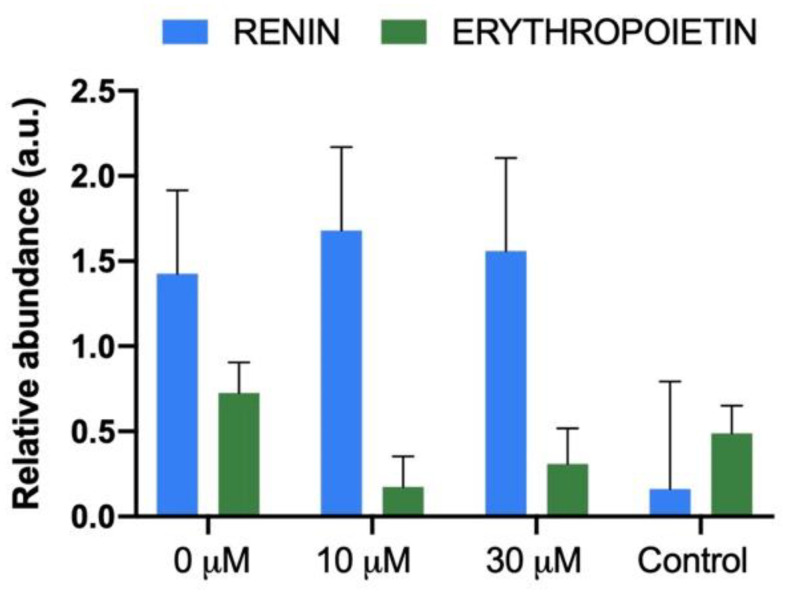
Renal gene expression levels of the renin and erythropoietin in the new kidney recovery 21 days after metanephroi allotransplantation embedding in sildenafil citrate (0, 10, and 30 µM), and the kidneys originated from a neonatal rabbit (1 week-old, control group).

**Table 1 jcm-11-03068-t001:** Primer sequences.

Genes	Sequence (5′-3′)	Product Size (bp)
REN	Forward: 5′-GGGACTCCTGCTGGTACTCT-3′	100
	Reverse: 5′-CTGAGGGCATTTTCTTGAGG-3′	
EPO	Forward: 5′-ACGTGGACAAGGCTGTCAGT-3′	162
	Reverse: 5′-TGGAGTAGATGCGGAAAAGC-3′	
GAPDH	Forward: 5′-GCCGCTTCTTCTCGTGCAG-3′	144
	Reverse: 5′-ATGGATCATTGATGGCGACAACAT-3′	

REN: renin; EPO: erythropoietin; GAPDH: glyceraldehyde-3-phosphate dehydrogenase.

**Table 2 jcm-11-03068-t002:** Histomorphometric quantification of the renal corpuscle of kidneys developed after metanephroi allotransplantation.

Sildenafil Citrate	*n*	Renal Corpuscle		Glomerulus		
		Area (μm^2^)	Perimeter (μm)	Area (μm^2^)	Perimeter (μm)	Cell Number
0 µM SC	10	3034.6± 176.44 ^b^	201.1 ± 6.06 ^b^	2132.5 ± 142.56 ^b^	170.9 ± 5.70 ^b^	41.0 ± 2.22 ^b^
10 µM SC	9	3639.7 ± 179.94 ^a^	218.5 ± 6.18 ^a^	2749.5 ± 145.39 ^a^	192.0 ± 5.81 ^a^	49.9 ± 2.26 ^a^
30 µM SC	8	3582.44 ± 187.59 ^a^	218.3 ± 6.45 ^a^	2655.7 ± 151.85 ^a^	190.2 ± 6.06 ^a^	48.3 ± 2.36 ^a^
Control	6	2633.4 ± 92.31 ^c^	184.2 ± 3.17 ^c^	2104.7 ± 74.58 ^b^	165.3 ± 2.98 ^b^	52.7 ± 1.16 ^a^

*n*: Number of new kidneys or control kidneys. Data are expressed as least-square means ± standard error of the mean. ^a,b,c^ Data in the same column with uncommon letters are different (*p* < 0.05).

## Data Availability

The datasets analysed during the current study are available from the corresponding author on reasonable request.

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
