# Peer review of "Sildenafil Citrate Enhances Renal Organogenesis Following Metanephroi Allotransplantation into Non-Immunosuppressed Hosts"

_jcm, 2022, doi:10.3390/jcm11113068_

Round 1

Reviewer 1 Report

Answers provided by the authors to some of the questions/concerns raised in the initial review comments are either incomplete or missing. Some important questions that should be addressed more thoroughly are as follows: 

  1. R3.6 Out of the total 7 host rabbits, how many rabbits received metanephroi treated with 0SC, or 10SC or 30SC? The exact number of rabbits used for each group is not specified in the Methods.
  2. R3.9 The authors state that a new figure was added for the wight of each new kidney obtained in 3.2, as per reviewer suggestion. However, no such figure can be found. The kidney weight is provided in the body of the text and not as a histogram. 
  3. A histological characterization of the nephron structures (staining for glomerular specific markers (e.g. nephrin, Ehd3) and tubule specific markers (e.g. CD10, slc3a1)) is lacking. This is important to demonstrate that the growing metanephroi are comprised of nephron structures and not random agglomeration of cells. 
  4. The normalization of the RT-PCR data is not clearly explained. It is not obvious what the expression data (y-axis) is relative to (one of these groups (control, 0SC group) would have been appropriate)?

Author Response

Answers provided by the authors to some of the questions/concerns raised in the initial review comments are either incomplete or missing. Some important questions that should be addressed more thoroughly are as follows: 

R3.6 Out of the total 7 host rabbits, how many rabbits received metanephroi treated with 0SC, or 10SC or 30SC? The exact number of rabbits used for each group is not specified in the Methods.

The exact number of recipients per group has been included in section 2.3. Metanephroi recovery and transplantation, line 137. Specifically, the sentence includes is "(2, 2 and 3 hosts for 0, 10 and 30 SC, respectively)."

R3.9 The authors state that a new figure was added for the wight of each new kidney obtained in 3.2, as per reviewer suggestion. However, no such figure can be found. The kidney weight is provided in the body of the text and not as a histogram. 

The figure was included in the version sent to Ms Lorina Ana Seras (Assistant Editor) when she asked us to respond to the reviewers who had reviewed the work many months earlier. Expressly, the information required is set out in Figure 4 below.

A histological characterization of the nephron structures (staining for glomerular specific markers (e.g. nephrin, Ehd3) and tubule specific markers (e.g. CD10, slc3a1)) is lacking. This is important to demonstrate that the growing metanephroi are comprised of nephron structures and not random agglomeration of cells. 

An immunofluorescence assay for detecting tubular integrity marker E-Cadherin has been included (Figure 6).

The normalization of the RT-PCR data is not clearly explained. It is not obvious what the expression data (y-axis) is relative to (one of these groups (control, 0SC group) would have been appropriate)?

The approach to gene expression could be assessed by relative or absolute quantification. The first one uses a standard curve from a pool with a known number of copies, and the second uses a standard curve from a pool with an unknown number of copies. When we wrote that it is relative, we do not indicate that we are using any group as a reference, only the methodology to calculate the gene expression.

Reviewer 2 Report

Remarks to the Author

The manuscript entitled “Sildenafil citrate enhances renal organogenesis following metanephroi allotransplantation into non-immunosuppressed hosts” has been re-reviewed. Obviously, a lot of careful work has gone into this project. This is an interesting and potentially important paper. However, as delineated in my review, a few issues must be addressed, including serious concerns about the purpose.

・Though the authors prepared Figure2, I do not think it is enough data to show the immunoresponse. If the rejection was occurred, what we first need to pay attention should be local reaction. The importance of this point was that the project was not used immunosuppressive drugs. Basically, we imagined a subtle reaction would occur. Whole counts of WBC in the blood may not imply the rejection of the graft until it induced infections. Please prepare some comments on this point in discussion.

・I agree with all three reviewers to prepare the quantitative data that show SC could enhance angiogenesis in this model. This should be most important point of this project to acknowledge Sidemafil citrate enhances renal organogenesis due to angiogenesis. If the Sidemafil citrate induced organogenesis without angiogenesis, I would retract the query.

・The order of Eosinophilis and Monocytes are correct? I am not sure, but C seemed monocytes. Am I wrong?

Author Response

The manuscript entitled “Sildenafil citrate enhances renal organogenesis following metanephroi allotransplantation into non-immunosuppressed hosts” has been re-reviewed. Obviously, a lot of careful work has gone into this project. This is an interesting and potentially important paper. However, as delineated in my review, a few issues must be addressed, including serious concerns about the purpose.

・Though the authors prepared Figure2, I do not think it is enough data to show the immunoresponse. If the rejection was occurred, what we first need to pay attention should be local reaction. The importance of this point was that the project was not used immunosuppressive drugs. Basically, we imagined a subtle reaction would occur. Whole counts of WBC in the blood may not imply the rejection of the graft until it induced infections. Please prepare some comments on this point in discussion.

We agree that our assessment has only focused on acute rejection, and on the systemic immune response as we did not administer immunosuppressants. This part is simply one more test that we systematically use, given that in all the work carried out both by us and previously by other authors, it has always been demonstrated that there is acute rejection when transplanting metanephros. Based on this review process, we consider the need to incorporate new immunological tests to assess the local reaction.

I agree with all three reviewers to prepare the quantitative data that show SC could enhance angiogenesis in this model. This should be most important point of this project to acknowledge Sidemafil citrate enhances renal organogenesis due to angiogenesis. If the Sidemafil citrate induced organogenesis without angiogenesis, I would retract the query.

Our study was designed to assess this capacity in the organ phenotype (organ weight), and this we understand has been demonstrated.  We only descriptively evidenced angiogenic activity with imaging, and given organ weight. These preliminary results open up future studies in which we will further investigate the angiogenic quantification of SC. We regret that we are unable to provide this quantitative information in this experimental design.

The order of Eosinophilis and Monocytes are correct? I am not sure, but C seemed monocytes. Am I wrong?

Oops! The position of the figures is correct, but the value of monocytes is in tens when they are units. The figure has been corrected. Thank you very much for catching this mistake.

This manuscript is a resubmission of an earlier submission. The following is a list of the peer review reports and author responses from that submission.

Round 1

Reviewer 1 Report

The authors presented a study of allogeneic transplantation of metanephroi in rabbits, with enhanced vascular response with the use of Sildenafil citrate. The methodology is mostly sound and the results suggest an interesting outcome. However, a few additional experiments would strengthen this paper and its' claims.

Major revisions:

  1. Please revise the manuscript for grammar and style.  It was difficult to read at times.
  2. Line 98 - Other aspects of immune response should be examined.  Not just lymphocyte count. Primarily I would include looking at the response in the implanted metanephroi for a local response.
  3. Figure 3 (but should be figure 4) - It appears that there may an increase in fibrotic response in the transplanted metanephroi based on histology images. It would be beneficial to include immunohistochemistry/fluoresence of other glomerulii and nephron markers and additionally injury markers or immune infiltration to indicate healthiness of the transplanted tissue in comparison to the healthy control. 
  4. If the authors are claiming that SC enhances glomerular filtration presumably due to neo-vascularization to the metanephroi.  However the data does not support this. Please include staining with PECAM1 at the very least to show that they are in fact vascularized.
  5. Figure 4 (which should be Figure 5) I don't believe RNA is the best way to analyze release of Renin and erythropoietin.  Just because RNA is present, it doesn't mean it is released in appropriate levels. I suggest including plasma measurements rather than RNA since that seems to be the standard method.  Also I'm not sure if this is a typo (line251), but gene expression from hepatic tissue should not used to measure renin and erythropoietin.  Please revise this figure.

Minor revisions:

  1. Line 53 - "without triggering an immune response"... Based on published papers, I don't think that the immune response was not triggered. It was significantly reduced. Please revise the sentence. 
  2. Were the metanephroi implanted into 2 week old host rabbits? Were the host kidneys from the same rabbits used for all comparisons? 
  3. Line 197 - what does "adult" mean?
  4. Figure 2 - Are the results mixed for all the SC conditions in this figure? It would be interesting and useful to see the counts for each SC condition separately to asses changes.
  5. Figure 3 B-D - What stage is this? Also there are 2x Figure 3, please correct.
  6. Table 2 - What's the significance in difference of cell number? Please explain in results and/or discussion.
  7. Line 258 - Saying "no immunological response" is too strong of a statement with just the lymphocyte measurements present. Please revise.
  8. I would not state that "our results support those of Hammerman's..." as it has not been shown conclusively in this paper. Also the citation is a review, not a study. Please revise.

Reviewer 2 Report

Dear,

I only have few comments listed below

  1. For the Figure 2, can the baseline data (prior to transplant) be added?  
  2. For the Figure 3 BCD, can the more quantitative analysis be included to show that SC can enhance angiogenesis in metanephrori?
  3. In the discussion, adding the mechanism of SC enhanced angiogenesis in terms of altered cell signaling pathways would be better.   

Reviewer 3 Report

Garcia-Dominguez et al describe their use of Sildenafil citrate (SC) to improve the enlargement of allotransplanted metanephroi. They transplanted embryonic stage metanephroi into adult rabbit hosts in 0uM, 10 uM and 30 uM SC and then waited for 3-weeks before the kidneys were retrieved and assessed. The authors suggest these retrieved kidneys are larger.

Whilst I liked the rationale for investigating SC, I feel there was insufficent rigor employed to conclusively prove that the SC is having specific effects on growth or engraftment in these kidneys.

Below are a few points that were of concern.

  1. Line 17: What does morphofutional mean? Should this be morphofunctional? I cannot find a definition for the former
  2. Line 20-22: The authors should include in the abstract that they performed allotransplantation after SC embedding and this improved engraftment?
  3. Line 34: This sentence isn’t coherent English
  4. Line 44: demandable should be demand
  5. How long are the kidneys in the SC droplets?
  6. What is the actual n number for the experiments? The authors show they used three rabbits as embryo donors and the 49 metanephroi were transplanted into 7 adult hosts (5-weeks old?), but there were three different sampling groups (0SC, 10SC and 30SC), so did they use only 2 or 3 hosts per sample? This should be clarified for the reader to judge the rigour of the experiments more clearly
  7. A recovery rate of 67% in the 0SC group seems to me to be a lot higher than 47% in the 30SC group, the authors should acknowledge this rather than say they are the same. This is important as it suggests SC might be having effects on the kidney that needs further investigation.
  8. The authors often refer to their figures as a whole, but not as the specific panels. For this reason, I have no real understanding for some of the data, for example why is Figure 3E shown? If it is as a comparison to the transplanted kidneys in Fig.3F-H then it should be at the same scale so they can be compared.
  9. The data given in 3.2 regarding renal weight should be given in the form of a histogram with all the individual weights plotted so the variation and deviation from the median can be inferred by the reader.
  10. Why wasn’t kidney weight to body weight ratio assessed? This normalizes the variation in the size of the host.
  11. Line 238 onwards is greatly overreaching. The authors give no evidence to support that the kidneys are producing urine or undergoing filtration activity through their glomeruli. It is worrying to me that they can be so conclusive in saying that the enlargement of the transplanted metanephroi “demonstrates the hydronephrotic state of the kidneys and their filtering activity”. This is a gross overstatement, there could be a whole raft of reasons as to why the kidneys got bigger other than urine production.
  12. Why was there no assessment of cell proliferation? Without this, how do the authors know for sure that it is growth? Hypertrophy could be involved also?
  13. What about nephron number? Surely this would suggest that nephrogenesis is proceeding, but the authors just look at the kidney as a whole rather than understand what is happening emprically.
  14. The authors suggestion that there is increased angiogenesis is very limited and not temporal. There is no assessment of angiogenesis other than an end-point showing of a blood vessel that the authors are assuming is a new vessel. Whilst it might be, it might also not be. Much more rigorous experiments are required before such strong conclusions can be made.
  15. Line 249: The authors say Figure 5 but they mean Figure 4.
  16. In 3.3 the authors say there are similar levels of renin and erythropoietin in experimental and control kidneys yet the data in Figure 4 is not consistent with this interpretation. Indeed, it isn’t clear to me how the authors have done the RT-PCR. In 2.6 they say that GAPDH was used as a normalizing gene, but I don’t know what the relative abundance is that they measure in the y-axis? This should be relative to 0 uM SC (or maybe the untreated control), but there are changes in these so I am at a loss as to what they are relating their PCR results against? Regardless, there are changes that would lead me to suggest that SC increases renin expression over controls.